# A Safety Analysis Method for Control Software in Coordination with FMEA and FTA

**Masakazu Takahashi** [1,*] **, Yunarso Anang** [2] **and Yoshimichi Watanabe** [1]

[1] Department of Computer Science and Engineering, University of Yamanashi, 400-8511 Kofu, Japan; nabe@yamanashi.ac.jp

[2] Department of Computational Statistics, Politeknik Statistika STIS, 13330 Jakarta, Indonesia; anang@stis.ac.id

* Correspondence: mtakahashi@yamanashi.ac.jp; Tel.: +81-55-220-8585

**Abstract:** In this study, we proposed a method to improve the safety level of control software (CSW) by managing the CSW's design information and safety analysis results, and combining failure mode and effects analysis (FMEA) and fault tree analysis (FTA). Here, the CSW is developed using structured analysis and design methodology. In the upper stage of the CSW's development process, as the input of the preliminary design information (data flow diagrams (DFDs) and control flow diagrams (CFDs)), the causes of undesirable events of the CSW are clarified by FMEA, and the countermeasures are reflected in the preliminary design information. In the lower stage of the CSW's development process, as the inputs of the detailed design information (DFDs and CFDs in the lower level) and programs, the causes of the specific undesirable event are clarified by FTA, and the countermeasures are reflected in the detailed design specifications and programs. The processes are repeated until the impact of undesirable events become the acceptable safety level. By applying the proposed method to the CSW installed into a communication control equipment on the space system, we clarified several undesirable events and adopted adequate countermeasures. Consequently, a safer CSW is developed by applying the proposed method.

**Keywords:** failure mode and effects analysis (FMEA); fault tree analysis (FTA); safety analysis; control software; structured analysis and design; software development

## 1. Introduction

Industrial products are controlled by a computerized system. The software installed into the control system is called control software (CSW). Therefore, these days, accidents caused by the CSW occur unexpectedly. In this paper, we propose a method for developing safer CSW. The characteristics of the proposed method are as follows: by maintaining the design information and the safety analysis results unitarily in the whole CSW's development process, one can develop safer CSW by conducting two kind of safety analysis. The proposed method contributes to realizing safer industrial products along with developing safer CSW.

First, we describe the background that the proposed method is required. Undesirable events of CSW have caused many industrial products troubles recently, such as an accident caused by the incorrect steering operation of an autonomous car [1], a loss of human lives caused by malfunction of radiation therapy equipment [2], and destruction of the spin satellite caused by abnormal hi-speed spin [3], resulting in enormous damages to human life and industrial products. Thus, a safe method for CSW is required. As the functional configuration of CSW becomes complex according to the high functionality of CSW, it becomes difficult to realize sufficient safety when just conducting individual safety analysis.

The proposed method adopts multiple safety analysis methods in adequate steps in the CSW development process, and the result of the safety analysis in a timely way reflects the design information. We propose a safer CSW development method by conducting

those tasks. In the upper stages of the CSW's development process (upper development process), we applied failure mode and effects analysis (FMEA) [4] to the CSW's design information, clarified the causes of undesirable events comprehensively, and adopted countermeasures for realizing safety. In the lower stages of the CSW's development process (lower development process), we applied fault tree analysis (FTA) [5] to the CSW's design information and code, clarified the causes of the specific undesirable events that could not be treated in the upper development process but were found in the use of CSW, and modified the design information and CSW. By repeating the design, development, FMEA, and FTA, we can clarify the undesirable events injected by the modifications and then identify those as modified (repaired). By executing the processes mentioned above, we can develop a safer CSW. In this way, the proposed method enables seamless sharing of design information and safety analysis results throughout the development process, and improves the safety of CSW by repeating designing and safety analysis. By developing safer CSWs, safer industrial products can be developed, and a safer society can be realized.

Here, we describe the CSW that is the target for this study. The target CSW has the following characteristics in the proposed method. The CSW is developed by applying structured analysis and design methodology (SADM) [6] and CSW is written in the C language. As discussed in Section 3, SADM is applied because the SADM and the V model of the software development process (this will be explained in Section 3.1) have high-affinity applications. On the other hand, the reason for assuming the C language is that the C language and SADM have high affinity, and 70% of CSWs are written in the C language. CSW works on a single chip, single-core Central Processing Unit (CPU). Those are based on the white paper of the embedded system development [7]. Large-scale CSW is realized by connecting small-scale control pieces of equipment via network and data bus. If small-scale CSW with low coupling and cohesion is developed, a safety analysis report for large-scale CSW is realized by gathering and combining the safety analysis reports of small-scale CSW. Thus, the proposed method can be applied to large-scale CSW.

The rest of the paper is organized as follows. Section 2 describes related works to the CSW's safety. Section 3 describes the proposed method's outline and the support environment. Section 4 describes the application and evaluation of the proposed method for actual CSW. Section 5 describes the conclusion and future works.

## 2. Related Works

We classified the related works as the established standards related to software safety, FMEA for software, FTA for software, other safety analysis methods, and cooperation with multiple safety analysis methods.

First, we describe the following established standards related to software safety. The International Electrotechnical Commission (IEC) 61,508 defined the requirements for the functional safety of programmable electric systems [8], recommending the V model of the software development process and requiring high coverage testing in checking the functions. In the automotive domain, the International Organization for Standardization (ISO) 26,262 provided a standard for a car's functional safety [9], which proposed a safety analysis method when considering undesirable events and improved CSW's safety by applying hazard and operability study (HAZOP) [10], FMEA, and FTA. IEC 82304-1 (health software: part 1), as the general requirements for product safety, was established as the standard of CSW's safety for medical equipment [11]. Good automated manufacturing practice ver. 5 (GAMP5) was established as the standard of CSW's safety for pharmaceutical production facilities [12]. These standards described the safety requirements for industrial products, analysis methods, and validation methods, but did not describe specific implementation methods. Therefore, we have to define the concrete procedures that met these standards.

Second, we explain related works for FMEA. Takahashi et al. [13] proposed an FMEA method of CSW for pharmaceutical production. This research made the execution of FMEA to the various kind of CSWs by defining the failure modes (failure modes will be explained in Section 3.2) in the unit of the function. Morita [14] proposed a method for

finding the bugs in the CSW by dividing the CSW into plural blocks, listing up the failure modes, and predicting the causes. Niwa [15] conducted countermeasures to improve CSW's reliability by listing up the function units' failure modes in the preliminary design specification. Goddard et al. [16] conducted FMEA by defining the failure modes in the CSW's instruction level. Snooke et al. [17] conducted FMEA by translating the CSW to the equivalent circuit. Lazarus et al. [18] proposed a failure analysis method by conducting FMEA to the class diagrams developed based on an object-oriented design methodology. Kim et al. [19] proposed a supporting method for detecting failure modes of various software installed into car equipment. Batbayar et al. [20] proposed a risk evaluation method for the CSW of medical equipment by combining a fuzzy model and FMEA. Yang et al. [21] developed a hybrid expert system of failure diagnosis for the embedded software by aggregating both case-based reasoning and diagnosis methods based on a Bayesian network. These researches proposed an FMEA method that focused on a narrow (specific) domain. This showed that the failure modes of these researches were not versatile. To resolve this problem, we have to develop failure modes applicable for many kinds of CSWs.

Third, we explain related work on FTA. Weber et al. [22] analyzed the cause of fault for avionics CSW written in assembler language by conducting FTA. Friedman et al. [23] proposed an automatic Fault Tree (FT) development method for software written in the Pascal language. Leveson et al. [24] prepared an FT template corresponding to the programming language instruction and developed an FT by combining those FT templates. Takahashi et al. [25] proposed an automatic FT development method by expanding Leveson's idea. Kumar et al. [26] showed that in the development of safety-critical ball position control systems, adequate CSW design could be achieved when the FTA was conducted in the software development life cycle's proper timing. Oveisi et al. [27] proposed a safety evaluation method by applying FTA to the software's sequence diagrams according to the object-oriented development methodology. Junga et al. [28] proposed an automatic FT development method from software specifications written in formal specification language called NuSCR. From the results of [24] and [25], we found that the causes of the undesirable events are detected at the program level by using FT. By improving this function, FTA for various types of CSWs becomes able to conduct.

Fourth, we explain other safety analysis methods. Hansen et al. [29] showed a method that detected the hazards comprehensively by applying the HAZOP to the software specifications written in the unified modeling language (UML). Guiochet et al. [30] proposed a method that clarified the hazard when the robot behaved by applying HAZOP to the robot's CSW specification written in UML. Kaleeswaran et al. [31] proposed a method to detect hazards written in domain-specific language by applying HAZOP if critical software was developed using model-based development. Abdulkhaleq et al. [32] clarified the system hazards of an adaptive cruise control system in the car by applying system-theoretic process analysis (STPA) and pointed out the constraint and problems of STPA application. Yang [33] proposed a method that derived the avionics system's safety validation testing requirements using STPA. Nakano et al. [34] showed that a safer system could be developed by clarifying crew return vehicles' hazards using STPA and reflecting the countermeasures corresponding to the hazard to CSW in the concept development phase. The completeness of the analytic result depends on the skills of the analyst because these methods have a high degree of freedom. Therefore, these methods are not suitable for use in the CSW's development.

Finally, we explain cooperation with multiple safety analysis methods. Hong et al. [35] proposed the following safety improving method: calculating the minimum cut-set of FT by conducting FTA, identifying the fault with the highest risk, and planning the countermeasure by conducting FMEA. Oveisi [36] proposed an approach to improve the CSW's safety throughout the software development lifecycle using hazard analysis in the upper development process, FTA and FMEA in the middle development process, and detailed

FTA and FMEA in the lower development process. From those cases, a safety analysis method coordinating with FMEA and FTA is valid because it has a high affinity.

Many related works on CSW have been conducted; however, no method in developing safer CSW in the whole development process has been reported. Our proposed method will be described in Section 3.

## 3. Outline of the Proposed Method

Here, we explain the outline of the proposed development method for a safer CSW. Section 3.1 explains the outline of the proposed method. Section 3.2 explains the FMEA for a CSW, analyzing causes of undesirable events comprehensively in the upper development process. Section 3.3 explains the FTA for CSW, analyzing the specific undesirable event's causes in the lower development process. Section 3.4 explains the safer CSW development environment by combining FMEA and FTA.

The technical term "fault" is defined as "the state that causes the degeneration or loss of ability for conducting a required function," and the technical term "failure" is defined as "the loss of ability for conducting a required function" (ISO/IEC 2382-14) [37]. Therefore, failure occurs when the fault occurs under specific conditions, resulting in undesirable events.

### 3.1. Outline of the Development Method for a Safer CSW

This subsection explains an analysis method for developing a safer CSW in the whole development process. The proposed method manages the CSW's design information and safety analysis results unitarily and uses that information by cooperating with FMEA and FTA. The safer CSW is developed by adequately reflecting the CSW's design information and safety analysis results.

First, we explain the development processes and the outputs. Figure 1 shows the CSW's development processes and outputs in each step. The CSW's development process shown in Figure 1 is called the V model of the software development process. The squares in Figure 1 show the individual processes. The left side of Figure 1 shows the development process, and the right side shows the verification process. The dotted line between the development process and the verification process shows the correspondence. We apply SADM to the development process of CSW. The upper development process consists of the planning step, requirement definition step, and preliminary design step. In the planning step, we define the outline of the CSW's functions and data interfaces (data and control signals sent and/or received from/to CSW; external entities (hardware, software, or persons)). The development plan is created as a document describing those data and control signals. In applying SADM, the documents corresponding to the development plan are the data context diagram (DCD) and control context diagram (CCD). In the requirement definition step, we define the requirements for CSW and create the requirement specification. Each requirement corresponds to the data flow diagrams (DFDs) process in the second and/or third layer. (DCD is considered as the first-layer DFD. DFDs are considered as the divided design of DCD. The number of the layer is a guide). We also define the large-grained functions. The preliminary design specifications are created as the document that describes large-grained functions. The large-grained functions correspond to the processes in the DFD of the third to fifth layer. After developing the DFD, control flow diagrams (CFDs) correspond to those DFDs, and control specifications (CSPECs) are developed consisting of state transition diagrams, decision tables, and activation tables. CFDs define the control signals used for invoking the processes and newly created control signals. The state transition diagrams define the inner state of the CSW, the transition conditions between the states, and the group of the processes activated when the transitions occur. The decision tables define the control signals issued by the ON/OFF combination of control signals. The activation tables define the groups of the processes activated by the control signals. In the lower development process, the large-grained functions are divided into several modules, and the algorithm of the module is defined. The detailed

design specifications are created as the modules' list and define the module's concrete processes. The modules correspond to the processes in the DFDs in the fourth to sixth layer. After developing the DFDs, CFDs and CSPECs corresponding to the DFDs are developed. Finally, in the programming step, CSWs are developed according to the detailed design specifications. Those items, such as DFDs, CFDs, and CSPECs, are registered into the database (DB), as explained in Section 3.4.

**Figure 1.** The control software (CSW)'s development process.

Next, we explain the safety analysis methods used in the CSW's development process. In the upper development process, the failures with the possibility of occurrence are examined comprehensively, and the necessary countermeasures are reflected in the designs. FMEA is used for the analysis of the failures and their causes. The details of the FMEA will be explained in Section 3.2. As the result of conducting FMEA, the development plan, the requirement specifications, and the preliminary design specifications shown on the left side of the area enclosed by the upper dotted bold line in Figure 1 are modified in relation to the failures considered to be critical. As a result of modifications, the negative impacts of failure are degraded to an acceptable level. Because the addition of the complex functions and the countermeasures on the program cannot be conducted in this step, the countermeasures are conducted in the lower development process, in which the countermeasures, such as the addition of complex functions and programs, are conducted. Additionally, the necessary countermeasures are conducted on the newly found faults. FTA is used for the analysis of the causes of the fault. FTA will be explained in Section 3.3. As shown in the left lower parts in Figure 1, the modification of the detailed design specifications, the modifications of the programs, and the addition of the programs are conducted according to the results of FTA. When the modifications of the detailed design specifications, the modification of the programs, and the addition of the programs affect the negative impact on the documents in the upper development process, the documents in the upper development process on the left side of the area enclosed by the lower dotted bold line in Figure 1 are modified.

By conducting the processes mentioned above, we have completed the first investigation of the CSW's safety. However, the possibility of injections of new failure modes and/or a fault occur in applying those countermeasures to the CSW is increased. Therefore, after completing the first investigation for the CSW's safety, FMEA and FTA are conducted to the CSW, and additional countermeasures are applied to the CSW. This investigation will repeat until the safety level of all failures and faults become acceptable. A safer CSW will be realized by those works.

*3.2. Outline of FMEA*

In this subsection, we explain the FMEA for investigating the CSW's safety in the upper development process.

First, we explain FMEA for CSWs. Here, we discuss the CSW's failure mode. Generally, a failure mode is defined as the change and/or alteration (violation) of the component in an adequate (healthy) state. As bugs were injected when developing the software, the component that contains the bugs is not in an adequate state originally. Therefore, the bugs are not considered as failure modes. The bugs should be removed by conducting the tests. The bug-removing task is not regarded as the target of FMEA in the proposed method. Accordingly, the failure modes dealt with in the proposed method are considered as follows: the deviation (violation) of usages of CSW functions (module, functions, etc.) and the deviations (violation) of the operation method. The difference between a bug and a failure mode is explained using a sample of stack overflow when an interruption occurs. Stack overflow occurs when many interruptions occur in the case that the values of the registers and variables pushed into the stack are not popped adequately, or when interruptions occur over the number that is defined in the requirement specification. As in the former case, when the gavages (the values of registers and variables) in the stack resulting from the inadequate software design are the cause, this is regarded as the bug. The cause of the latter case is too much push operation. In this case, the software is designed adequately, but more interruptions than expected occur. Because this is a violation of the requirement, this is regarded as a failure mode (violation of software usage). The benefit that violation of the usage and operation is regarded as the failure mode is that these failure modes can apply to the various types of function, and the number of failure modes can be reduced. If the bugs were regarded as failure modes, the number of failure modes would be enormous because the types of the bugs are various. In the proposed method, the common failure modes used for whole CSWs and standard countermeasures for each common failure mode are derived by analyzing more than twenty existing CSWs [10]. Table 1 shows a list of common failure modes and standard countermeasures. Here, a sample of common failure modes are explained. The common failure mode regarding the start-up (execution) function is that the conditions for executing the function are not satisfied, and when this situation occurs, the function becomes unable to execute. The countermeasures for this problem are considered as follows: add the execution-condition check to the standard operation procedure (SOP), and add the function that sets the conditions.

**Table 1.** Common failure modes and standard countermeasures.

| Group | Common Failure Mode | Failure Example | Countermeasure Policy | Standard Countermeasures |
|---|---|---|---|---|
| Startup | The startup conditions for functions are not prepared | Related operations cannot be conducted, an improper system status exists | Review the startup conditions | Add the confirmation procedure for the startup conditions to the Standard Operation Procedure (SOP), set the conditions as to whether or not to start |
| | | | Conduct multiple checks when startup | |
| | | | Conduct the startup check | Display the startup status |
| Termination | The termination conditions for functions are not prepared | Related operations cannot be conducted, an improper system status exists | Review the termination conditions for functions | Add the confirmation procedure for termination conditions to the SOP, set the conditions whether or not to terminate, multiplex the termination confirmation procedure |
| | | | Conduct multiple checks upon termination | |
| | | | Conduct termination check | Display the termination status |
| | | | Transit to the safe status for top priority | Add the emergency stop function |

**Table 1.** *Cont.*

| Group | Common Failure Mode | Failure Example | Countermeasure Policy | Standard Countermeasures |
|---|---|---|---|---|
| Input/Output | Instructions on SOP misread | Improper results are calculated, an improper system status exists | Conduct multiple checks on SOP | Conduct double checks on SOP |
| | | | Improve the visibility of SOP indications | Integrate the SOP format |
| | Indications on Human Machine Interface (HMI) misread | Improper results are calculated, an improper system status exists | Conduct multiple checks on HMI | Conduct double checks on HMI |
| | | | Improve the visibility of HMI | Integrate the HMI format |
| | | | Check the content of HMI | Add the reconfirmation function |
| | Mistake in checking products | Improper results are calculated | Conduct multiple checks on products | Conduct double checks on products |
| | Past data are lost | Data related to quality are lost | Notify when data are lost | Add a warning function for past data loss |
| | Latest data are lost | | Notify if there is a data loss risk | Add a warning function for the latest data loss |
| | An inputting error | Improper results are calculated, an improper system status exists | Multiple checks on input data | Conduct double checks on setting data |
| Calibration | Long time intervals for function calibration | A wrong measurement is done, improper results are calculated | Conduct periodic reviews | Shorten time intervals for function calibration |
| Qualification | Wrong operation authority | Proper operations cannot be done, improper results are calculated | Confirm the qualification before operation | Confirm operation authority before operation |
| | | | Do not set improper authority | Review authority periodically |
| Backup | Insufficient backup | Data disappear, data related to quality are lost | Conduct proper backup operations | Organize the backup procedure in the SOP |
| | | | Shorten backup intervals | Shorten backup time intervals |
| Unexpected CPU Load | Unexpected data update occurs | Data cannot be updated | Realize faster processing | Realize faster update processing |
| | | | Develop faster devices | Install faster memory devices |
| | The upper limit of calculation precision is confirmed | Improper results are calculated | Increase significant digits | Utilize double-precision variables |
| | The lower limit of calculation precision is confirmed | Improper results are calculated | Increase significant digits | Utilize double-precision variables |
| | Divided by zero | Operation is suspended | Give a warning of division by zero | Add a warning function for a small divisor |
| | Unexpected amount of data is accepted | Abnormal program shutdown | Refuse data | Add a restriction function for available data |
| | | | Do not input data | Add the number of available data to the SOP |
| | Unexpected interruption tasks occur | | Restrict interruption tasks | Restrict interruption tasks |
| | | | Prohibit interruption tasks | Add the restriction function for interruption tasks to the SOP |
| | Unexpected CPU load occurs | Program does not response, a slow response | Unexpected execution requests are not sent | Add the function of displaying CPU usage |
| | | | Refuse unexpected execution requests | Add the restriction function for accepting execution requests under CPU overload |

**Table 1.** *Cont.*

| Group | Common Failure Mode | Failure Example | Countermeasure Policy | Standard Countermeasures |
|---|---|---|---|---|
| Malicious operations or attacks | No identification for important data | Data are removed | Take measures so that data are not removed | Introduce Data Loss Prevention (DLP) tools |
| | No access control for data | | Take measures so that data are not accessed | Add access control for data according to each user |
| | Data could be rewritten | Data are falsified | Take measures so that data are not falsified | Add e-signature, add time stamp |
| | Vast amounts of data sent | Related operations cannot be conducted | Data acceptance is blocked | Disconnect from the external network |
| | Vast amounts of requests sent | | Data are selected | Install fire walls |
| | Illegally accessed from the outside | System is invaded | Disconnect | Disconnect from the external network |
| | | | Discover illegal access | Introduce Intrusion Detection System (IDS), Introduce Intrusion Prevention System (IPS) |
| | Data with virus attached are received | System malfunctions, improper results are calculated | Remove computer virus | Introduce antivirus software |
| | | | Take measures so that virus does not invade | Introduce virus protection software |
| | | | | Conduct virus check on USB memory devices connected |

Figure 2 shows the process of conducting the proposed FMEA. First, the CSW is designed, and the requirement specifications and preliminary design specifications are registered to the Design Safety Information DB, as described in Section 3.4. Second, the CSW's functions are extracted from the DB, and the common failure modes shown in Table 1 are investigated as to whether they apply or not to the extracted functions. Third, if the common failure mode can apply to the function, the function–failure mode correspondence table is developed. Fourth, the function, the common failure mode, and the negative impact for the CSW are decided using the FMEA sheet (Table 2). The usage of the FMEA sheet is as follows: the function is described (entered) on the 1st column of the sheet, the common failure mode applicable to the function is clarified and described on the 2nd column of the sheet, and the impact on the system when the common failure mode occurs is clarified and described on the 3rd column of the sheet. Then, the method described below is used to determine whether the effect is acceptable. Fifth, severity, incidence, risk class, and detection rate for the failure mode unit are decided, and the risk priority is calculated. When deciding the risk priority, the risk evaluation matrix is used. Figure 3 shows the risk evaluation matrix. On the left side of Figure 3, the value of the risk class is calculated using severity and incidence. On the right side of Figure 3, the value of the risk priority is calculated using risk class and detection rate. Sixth, based on the value of the risk priority, it is judged whether the CSW can accept the failure or not. If the risk cannot be accepted, it is investigated to apply the standard countermeasures described in Table 1. Finally, the severity, incidence, risk class, and detection rate when applying the standard countermeasures are evaluated, and the risk priority is calculated again. As a result, if the CSW can accept the failure, the CSW's safe tasks are completed using FMEA. If the CSW cannot accept the failure, the same tasks are repeated until CSW can accept the failure.

List of Common Failure Mode

| Group | Common Failure Mode | Failure Example | Countermeasure Policy | Standard Countermeasures |
|---|---|---|---|---|
| Startup | The startup conditions for functions are not prepared | Related operations cannot be conducted, an improper system status exists | Review the startup conditions | Add the confirmation procedure for the startup conditions to the SOP, set the conditions as to whether or not to start |
| | | | Conduct multiple checks when startup | |
| | | | Conduct the startup check | Display the startup status |
| Termination | The termination conditions for functions are not prepared | Related operations cannot be conducted, an improper system status exists | Review the termination conditions for functions | Add the confirmation procedure for termination conditions to the SOP, set the conditions whether or not to terminate, multiplex the termination confirmation procedure |
| | | | Conduct multiple checks upon termination | |
| | | | Conduct termination check | Display the termination status |
| | | | Transit to the safe status for top priority | Add the emergency stop function |
| - - - - | - - - - - | - - - - - | - - - - - | - - - - - |

refer & use

Function - Failure mode Correspondence Table

function1
function2
- - - -

Design Safety
information DB

input

| Requirement Spec. / Preliminary Design Spec. | Gropu | Common Falure Modes |
|---|---|---|
| Function A | startup | The startup conditions for the function are not prepared. |
| | termination | The termination conditions for the function are not prepared. |
| | input/output | Instruction on SOP missed. |
| Function B | startup | The startup conditions for the function are not prepared. |
| | input/output | Instruction on SOP missed. |
| | input/output | Mistake in checking product. |
| - - - - | - - - - | - - - - |

Analyze correspondences between functions and common failure modes considering similarity.

develop

Result of Failure Mode and Effects Analysis using FMEA Sheet

| Function | Common Failure Mode | Impact to System | Accept /Reject | Severity | Incidence | Risk Class | Detection rate | Priority | Countermeasures |
|---|---|---|---|---|---|---|---|---|---|
| Function A | Startup condition X is not prepared. | The machine does not work. | ○ | Middle | Low | 3 | High | Low | Add SOP for Checking Startup Conditions . |
| | | | | Middle | Low | 3 | High | Low | |
| | Termination condition Y is not prepared. | The machine use electric power continously. | ○ | Middle | Low | 3 | High | Low | Add SOP for cheking termination Condition. |
| | | | | Middle | Low | 3 | High | Low | |
| - - - - | - - - - | - - - - | -- | -- | -- | -- | -- | -- | - - - - |

**Figure 2.** The procedure of the proposed failure mode and effects analysis (FMEA).

**Table 2.** Sample of failure mode and effects analysis (FMEA) sheet.

| Function | Common Failure Mode | Impact to System | Accept/Reject | Severity | Incidence | Risk Class | Detection Rate | Priority | Countermeasures |
|---|---|---|---|---|---|---|---|---|---|
| Function A | Startup condition X is not prepared. | The machine does not work. | Accept | Middle | Low | 3 | High | Low | Add Standard Operation Procedure (SOP) for Checking Startup Conditions. |
| | | | | Middle | Low | 3 | High | Low | |
| | Termination condition Y is not prepared. | The machine use electric power continuously. | Accept | Middle | Low | 3 | High | Low | Add SOP for checking termination Condition. |
| | | | | Middle | Low | 3 | High | Low | |
| - - - - | - - - - | - - - - | – | – | – | – | – | – | - - - - |

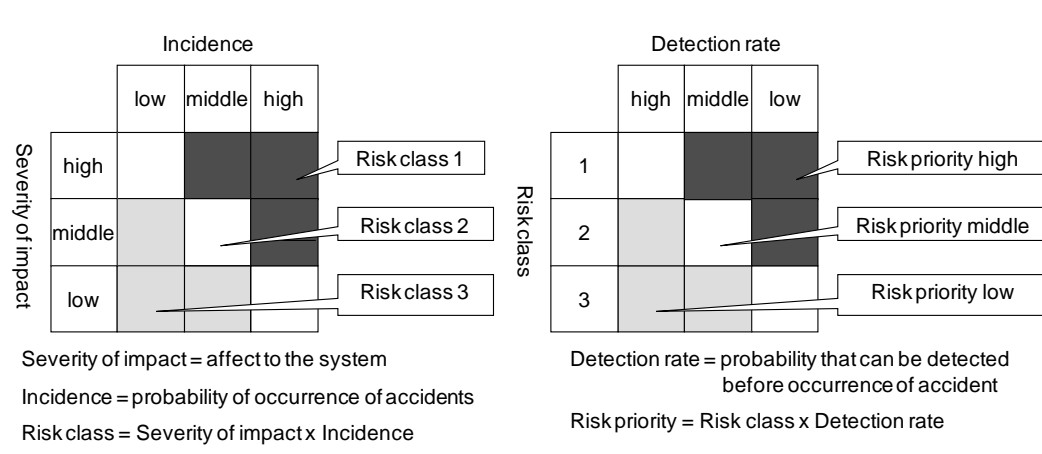

Severity of impact = affect to the system

Incidence = probability of occurrence of accidents

Risk class = Severity of impact x Incidence

Detection rate = probability that can be detected before occurrence of accident

Risk priority = Risk class x Detection rate

**Figure 3.** Risk evaluation matrix.

### 3.3. Outline of FTA

This subsection describes the FTA used for clarifying the specific causes of the CSW's fault in the lower development processes.

First, FTA for CSW is explained [38]. The components of CSW are instructions, and the interfaces to other components are the execution sequence of the instructions and the data exchanged. The fault is propagated via the sent/received data between the instructions according to the instruction's execution sequence. Therefore, to identify the causes of the fault, we need to trace the execution sequence of instructions and calculation process in the opposite direction. We explain this reason using the sample program shown in Figure 4. The function of this program is to input the initial value and add the value of the index in the for-loop to the initial value five times. We consider a case that the value of variable res1 in line 07 becomes six (this is a focused result). The instruction in line 05 is executed before the execution of the instruction in line 07, and the value of res1 is five before the execution of the instruction in line 05. As the instruction in line 05 exists in the loop instruction in line 04, the instruction is executed five times. The value of res1 is one before executing the loop instruction in line 04. The instruction in line 02 is executed before the execution of line 04, and the value of one is assigned into res1. As the result that the value of one is assigned into res1 in line 02, we found that res1 becomes six in line 07. To clarify the calculation process between the instruction executing before and after, the relationship (calculation process) between the instruction executing before and after is defined as the Fault Tree template (FT template; explained later). Here, because it is inefficient to develop FT templates for all instructions of the C language, FT templates for the instructions that are used frequently are developed based on the result of instruction appearance frequency in existing CSWs. The FT is developed by clarifying the instruction that is executed before the focused instruction and combining the FT templates corresponding to the instruction. This procedure is defined as FT development rules (explained later).

Figure 5 shows the proposed FTA procedure. In the proposed method, first, the CSW's design information (processes in the fourth and/or fifth layer's DFDs, CSPECs, codes, etc.), fault, and instruction that causes the fault are obtained from the DB, as described in Section 3.4. Second, it extracts the FT template corresponding to the instruction that causes the fault from the FT template library. Third, the FT template calculation is modified according to the CSW's execution status, and the FT template is regarded as the temporary FT. Fourth, the instruction executed immediately before is clarified according to the FT development rules; the corresponding FT template is extracted from the FT template library; the corresponding FT template is added (combined) to the temporary FT; the contents of the temporary FT are modified according to the actual status of the CSW. By conducting those tasks, we can develop the FT for the fault.

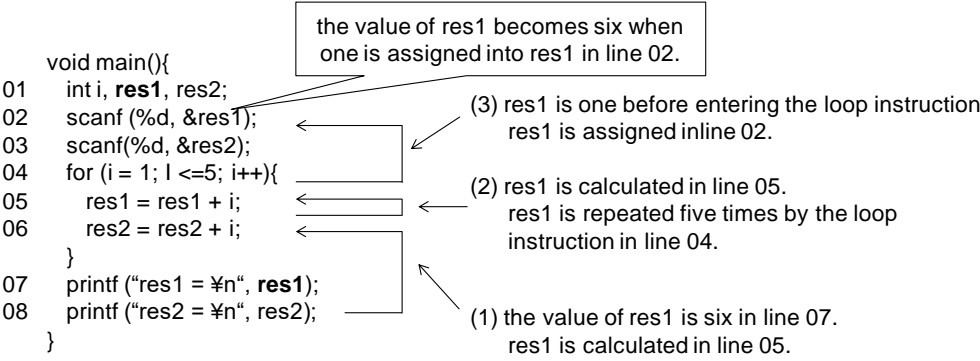

**Figure 4.** Sample program for explanation of the proposed method.

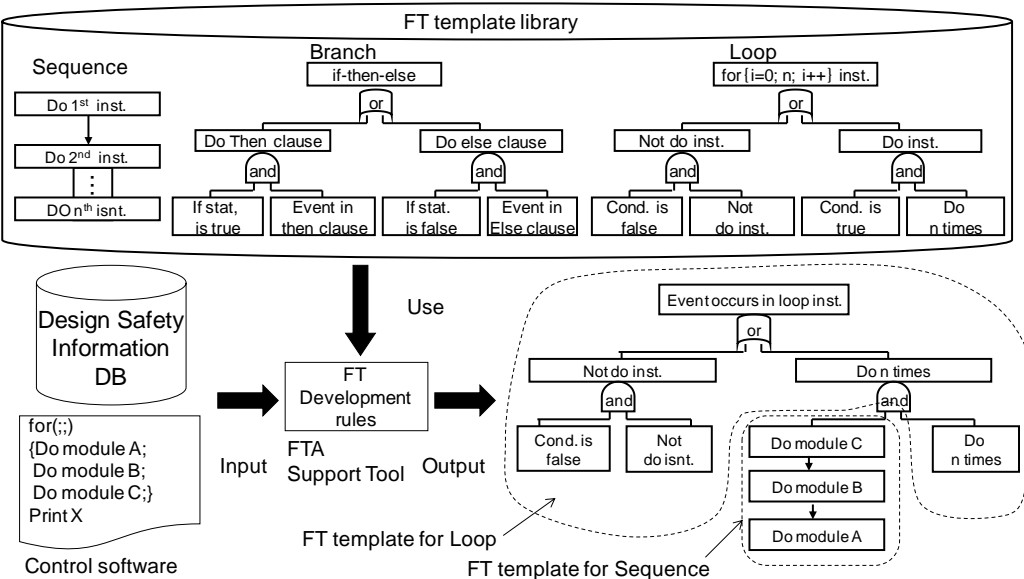

**Figure 5.** The procedure of the proposed fault tree analysis (FTA).

Second, we explain the FT templates. In the proposed method, we develop the FT templates. The FT templates are developed based on the result of the appearance frequency of the instructions in the existing twenty CSWs. Figure 6a–i shows each FT template. These FT templates are described using logical symbols. In each FT template, it only shows the relationship between the event after the execution of an instruction (TEvent) and the event before the execution of that instruction (BEvent). Figure 6a shows the FT template for the assigned statement. This template explains that the causes of the undesirable event are the case of "input value is incorrect" or the case of "operator is incorrect" when this instruction is executed. Figure 6b shows the FT template for the if-then-else statement. This template explains that the causes of the undesirable event are the case that "n-th clause is executed and the instruction in the clause produces the event" or the case that "else clause is executed and the instruction in the clause produces the event" when this instruction is executed. Figure 6c shows the FT template for the whole statement. This template explains that the causes of the undesirable event are the case that "as a result of not executing this instruction, the undesirable events occur" or the case that "as a result of executing this instruction n-th times, the undesirable event occurs" when this instruction is executed. Figure 6d shows the FT template for the function call. This template explains that the cause of the undesirable event is the case that "arguments are incorrect" or the case that "the function cannot be executed" when this instruction is executed. Figure 6e shows the FT template for interruption. This template explains that the causes of the undesirable event are the case that "the interruption occurs and the interruption module produces the undesirable event", the case that "the interruption does not occur, and the none-execution of the interruption module produces the undesirable event", or the case that "the inhibition of the interrupt produces the undesirable event". Figure 6f shows the FT template for the global variables. This template explains that the cause of the undesirable event is the case that "one or more values of the global-variables in the whole program being incorrect produce the undesirable event". Figure 6g shows the FT template for the local variables. This template explains that the cause of the undesirable event is that "one or more values of local variables in the focused scope being incorrect produce the undesirable event". Figure 6h shows the FT template for the array. This template explains the cause of the undesirable event is the case that "the index of an array is under zero", the case that "the element that the index points to does not exist", or the case that "the value of the element that is pointed to by the index is incorrect". Figure 6i shows the template for the pointer. This template explains that the cause of the undesirable event is the case that "the address does not exist", the case that "the address is incorrect", or the case that

"the stored value is incorrect". Here, the instruction that has the hierarchical instruction structure is called the hierarchical instruction. The FT template for hierarchical instruction is developed by combining the existing FT templates. When a frequently used instruction appears, the instruction's FT template is developed and added to the FT template library. Since the FT template only shows the relationship between before and after the execution of an instruction, there is no negative impact on the other FT template and FT development rules by adding a new FT template.

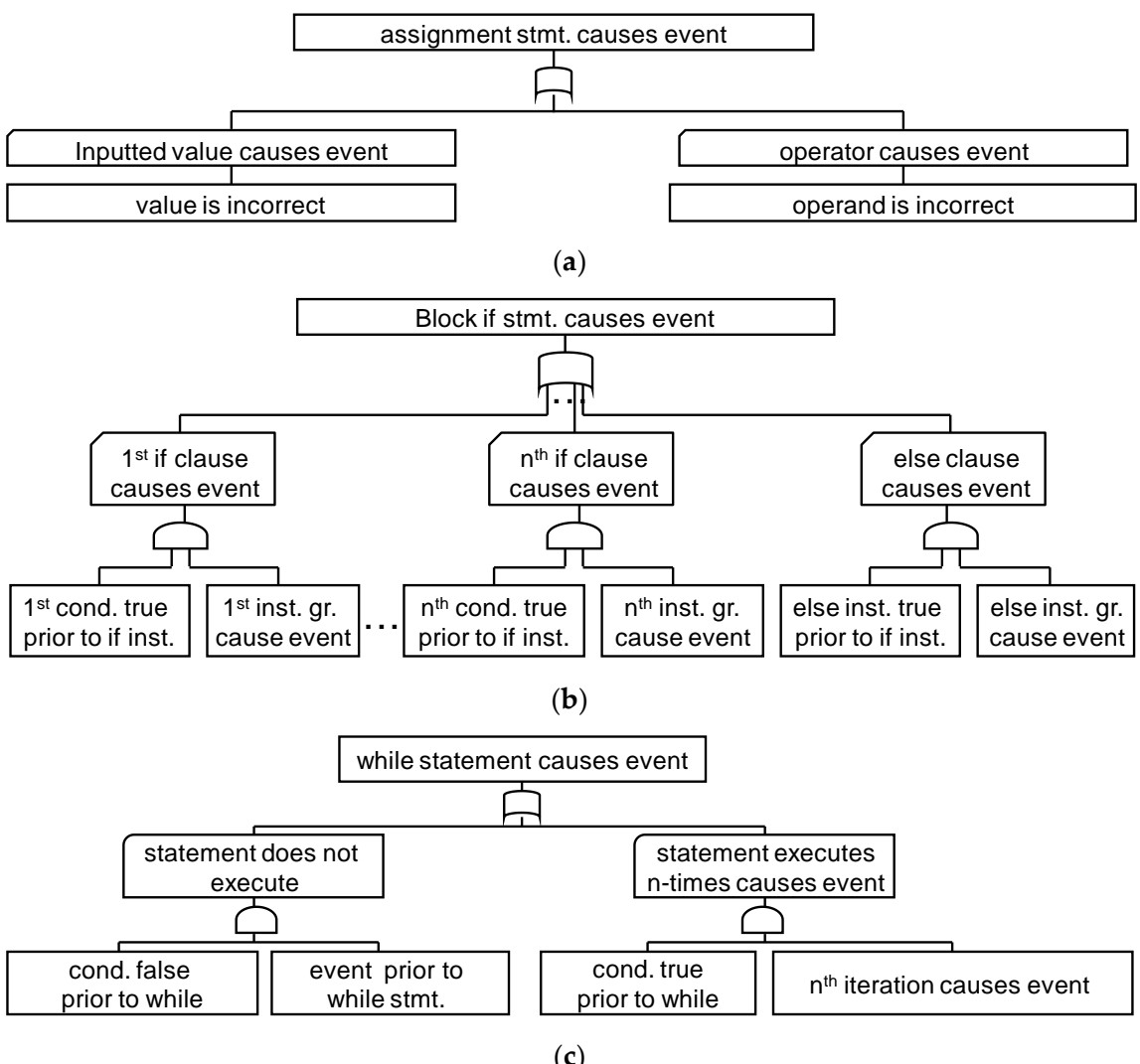

**Figure 6.** *Cont.*

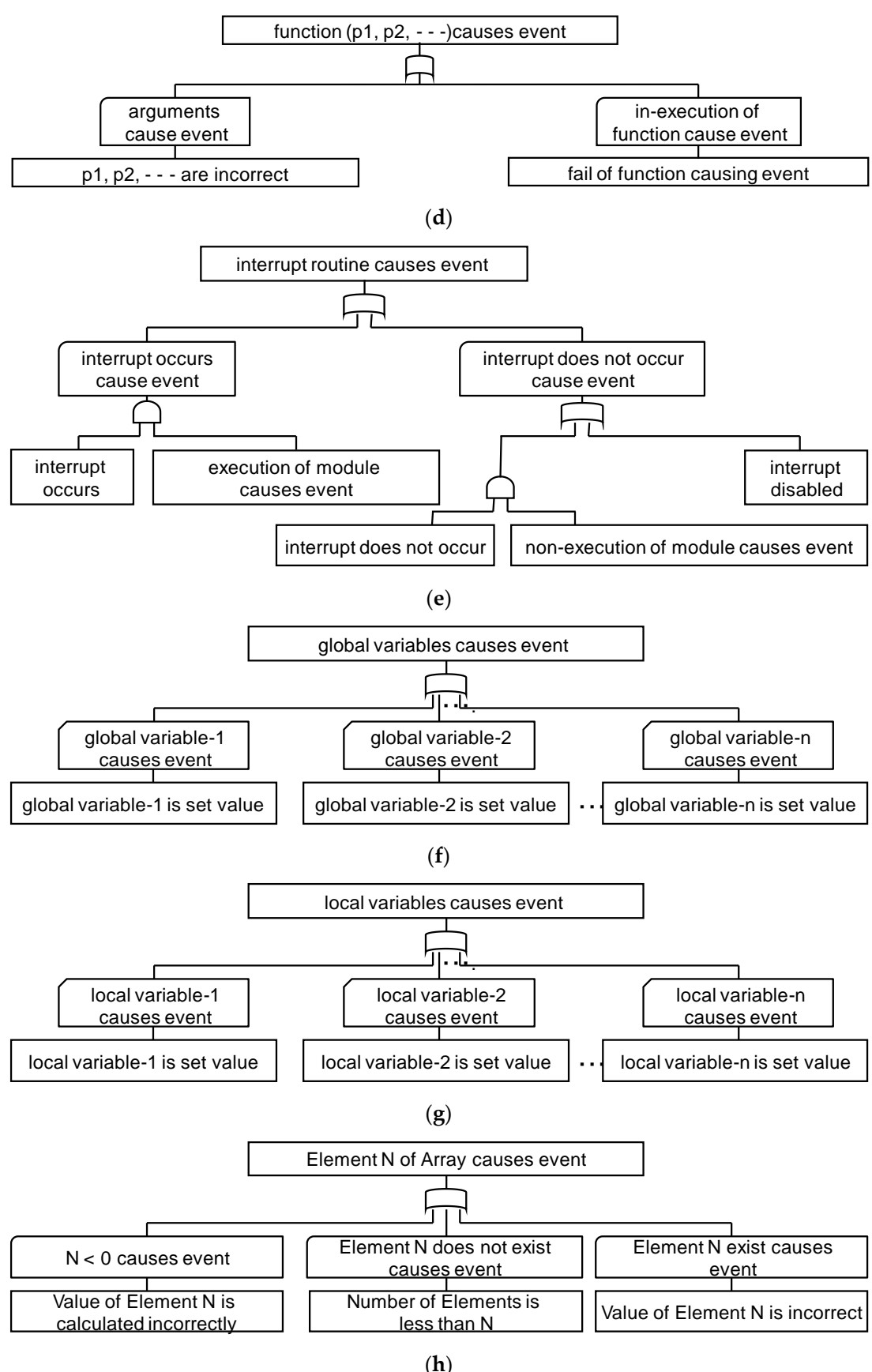

**Figure 6.** *Cont.*

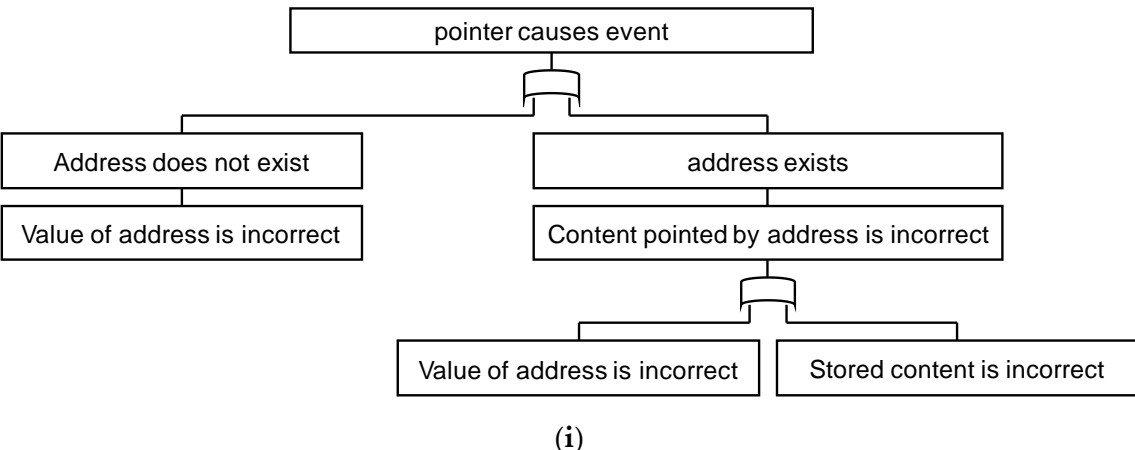

(**i**)

**Figure 6.** (**a**) Fault tree (FT) template for the Assigned Statement. (**b**) FT template for the If-Then-Else Statement. (**c**) FT template for the While Statement. (**d**) FT template for the Function Call. (**e**) FT template for the Interrupt. (**f**) FT template for the Global Variables. (**g**) FT template for the Local Variables. (**h**) FT template for the Array. (**i**) FT template for the Pointer.

Third, we explain the FT development rules. Figure 7a–d shows the FT development rules, and Figure 7a shows the outline of the FT development rules. Here, dFT means developing FT (i.e., the FT to be targeted), and wFT means the temporary FT (under-working FT) used in the "Develop FT" module. As shown in Figure 7a, FT development rules set the fault to the top event in dFT, and the "Develop FT" module develops the remaining part of dFT. The "Develop FT" module clarifies the instruction executed immediately before executing the instruction and combines the corresponding FT template. The tasks are repeated until it is no longer possible to be back in the execution order of the instructions. Finally, the instruction that exists at the forefront of the FT is the fault. Next, the contents of the "Develop FT" module are explained. Figure 7b shows the modules consisting of the "Develop FT" module. The "Develop FT" module develops the dFT corresponding to the $I_g$ contained in the BEvent of dFT. Additionally, the "develop wFT corresponding to instruction ($I_x$)" module shows the wFT development procedure corresponding to the instruction $I_x$; the "operation for global variables" module shows the procedure detecting global variables in BEvent and adding an FT template for global variables to the dFT; the "operation for local variables" module shows the procedure detecting local variables in BEvent and adding an FT template for local variables to the dFT. Figure 7c shows the outline of the "2.1 develop dFT when $I_g$ is a hierarchical instruction", whereas Figure 7d shows the outline of the "2.2 develop dFT when $I_g$ is not a hierarchical instruction". By combining the FT template according to the reverse execution sequence from the instruction that causes the fault to the instruction that makes the fault occur, we clarify the causes of the fault.

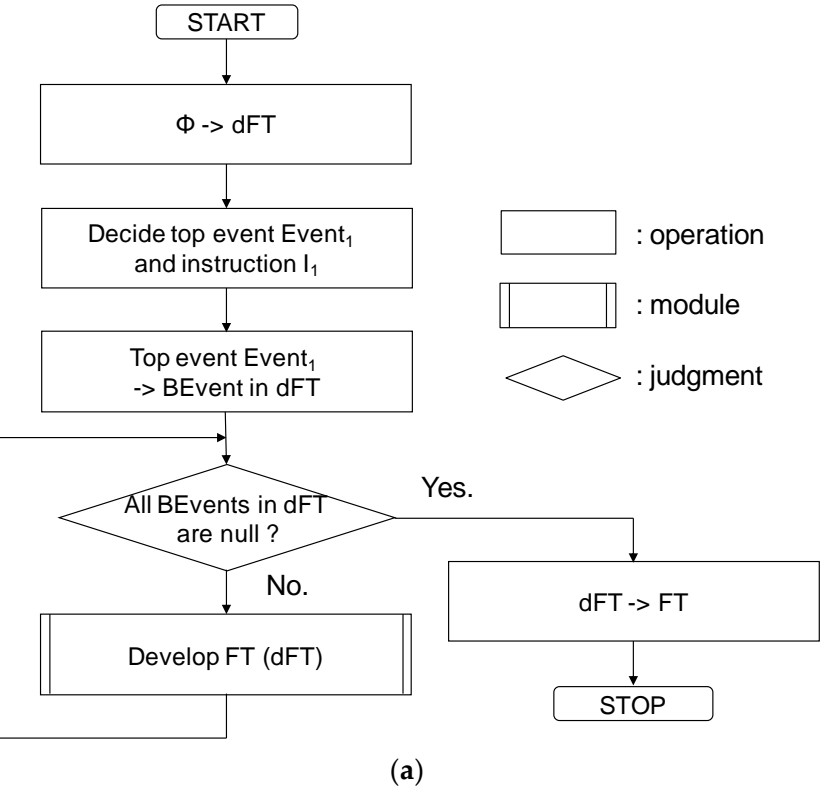

(**a**)

Develop FT(dFT)
1. initialize wFT.
2. develop dFT corresponding to $I_g$ included in all BEvents.
  2.1. develop dFT when $I_g$ is hierarchical instruction.
  2.2. develop dFT when $I_g$ is not hierarchical instruction.
3. return dFT

develop wFT corresponding to instruction ($I_x$) : develop wFT for $I_x$
(1) regard FT template corresponding to Ix as wFT.
(2) set BEvent in dFT to TEvent in wFT.
(3) calculate all BEvents in wFT.

operation for global variables: development of dFT when global variables are included in BEvents
(1) FT that is added FTT for global variables to BEvent in wFT regards new wFT
(2) write all BEvent in wFT
(3) combine dFT to wFT

operation for local variables: development of dFT when local variables are included in BEvents
(1) FT that is added FTT for local variables to BEvent in wFT regards new wFT
(2) write all BEvent in wFT
(3) combine dFT to wFT

(**b**)

**Figure 7.** *Cont.*

2.1. develop dFT when $I_g$ is hierarchical instruction.
(1) obtain hierarchical instruction that $I_g$ belongs, and get maximum layer $L_{max}$ and layer L that $I_g$ belongs
(2) k = Lmax
(3) Develop dFT at the layer that hierarchical instruction $I_g$
　(3-1) develop wFT corresponding to instruction ($I_g$)
　(3-2)　following tasks are conducted for all BEvent in wFT
　(3-2-1) when global variables are contained in BEvent
　　conduct operation of global variables, and k = k - 1
　(3-3-2) when local variables are contained in BEvent
　　conduct operation of local variables, and k = k - 1
　(3-2-3) when global variables and local variables are contained in BEvent
　　conduct operation of global variables and local variables, and k = k - 1
(4)　Develop dFT at except hierarchical instruction that $I_g$ belongs
　(4-1) when L < k < Lmax
　　(4-1-1) develop wFT corresponding to instruction (instruction $I_l$ at k-th layer)
　　(4-1-2) following tasks are conducted for BEvent in wFT
　　　(4-1-2-1) when global variables are contained in BEvent
　　　　conduct operation of global variables, k = k − 1, and go to (4)
　　　(4-1-2-2) when local variables are contained in BEvent
　　　　conduct operation of local variables, k = k - 1, and go to (4)
　　　(4-1-2-3) when global variables and local variables are contained in BEvent
　　　　conduct operation of global variables and local variables, k = k - 1, and go to (4)
　(4-2) when k = L
　　(4-2-1) develop wFT corresponding to instruction (instruction $I_i$ at k-th layer)
　　(4-2-2) following tasks are conducted for BEvent in wFT
　　　(4-2-2-1) when global variables are contained in BEvent
　　　　conduct operation of global variables, and k = 0
　　　(4-2-2-2) when local variables are contained in BEvent
　　　　conduct operation of local variables, and k = 0
　　　(4-2-2-3) when global variables and local variables are contained in BEvent
　　　　conduct operation of global variables and local variables, and k = 0

(**c**)

2.2. develop dFT when $I_g$ is not hierarchical instruction.
(1) develop wFT corresponding to instruction ($I_g$)
(2) conduct following tasks for all BEvents in wFT
　(2-1) when global variables are contained in BEvent
　　conduct operation of global variables
　(2-2)when local variables are contained in BEvent
　　conduct operation of local variables
　(2-3) when global and local variables are contained in BEvent
　　conduct operation of global variables and local variables

(**d**)

**Figure 7.** (**a**) FT development rules—Outline. (**b**) FT development rules—Detail of Develop FT module. (**c**) FT development rules—Detail of "Develop dFT" when $I_g$ is hierarchical. (**d**) FT development rules.—Details of "Develop dFT" when $I_g$ is not hierarchical.

*3.4. Safety Analysis Support Environment*

This subsection explains the safety analysis support environment combined with the contents described in Sections 3.1–3.4.

Figure 8 shows an outline of the proposed safety analysis support environment, consisting of the FMEA support tool, the FTA support tool, and the design and safety information database (DSDB).

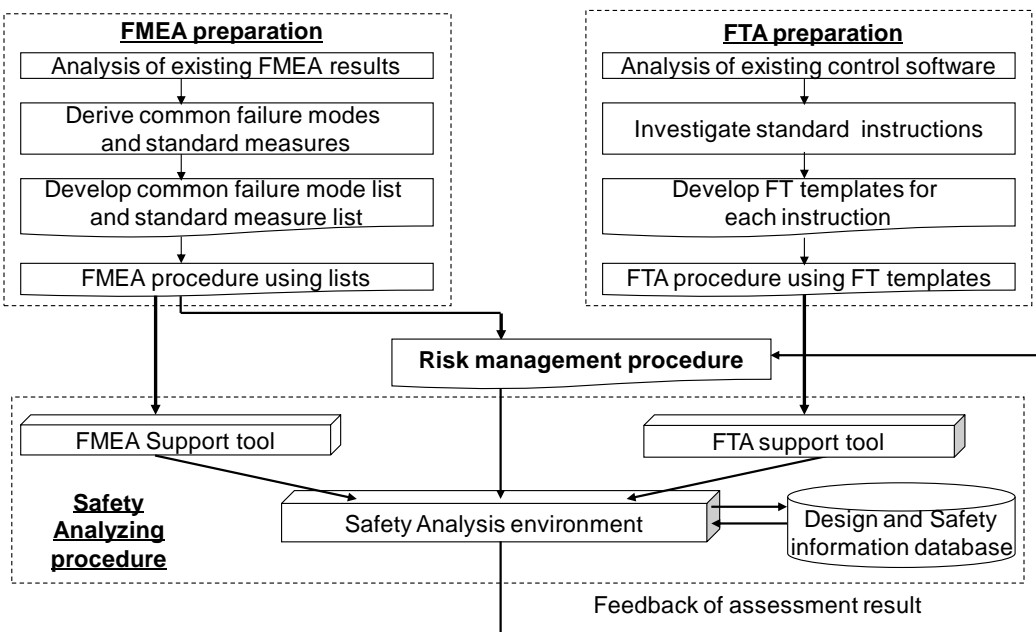

**Figure 8.** Outline of the proposed safety analysis support environment.

The FMEA support tool is used in developing the requirement specifications and the preliminary design specifications in Figure 1. The inputs of the FMEA support tool are all functions extracted from DSDB. The outputs of the FMEA support tool are the results of FMEA.

The FTA support tool is used in the step for developing the detailed design specifications and program in Figure 1. The FTA support tool's inputs are the fault and the CSW's design information extracted from DSDB. The faults are items that cannot be modified when FMEA is conducted and items clarified after completing the CSW development. The outputs of the FTA support tool are the FTs for the fault.

DSDB manages the data necessary for conducting FMEA and FTA. Figure 9 shows the outline of the DSDB data structure. DSDB consists of the following tables: requirement specification, preliminary design specification, detailed design specification, module, common failure mode and countermeasure, function–common failure mode correspondence, actual failure and impact, an actual countermeasure for failure, fault, module–fault correspondence, fault tree, and actual countermeasure for fault. Those tables are created in the unit of the CSW's version. The tables for each specification include information about CSW's functions. The module table maintains the source code of the CSW. The common failure mode and countermeasure table include all common failure modes and all countermeasures for the failure. The function–common failure mode correspondence table manages all common failure modes applied to each function. The actual failure and impact table includes concrete failure and its negative impact. The actual countermeasure for failure table manages the actual countermeasures applied to the CSW. The fault table registers all investigated faults. The module–fault correspondence table manages the information related to the actual conduction of FTA. The fault tree table manages the result of FTA (FTs) calculated by the FTA support tool. The actual countermeasure for the fault table manages the concrete countermeasures for the fault.

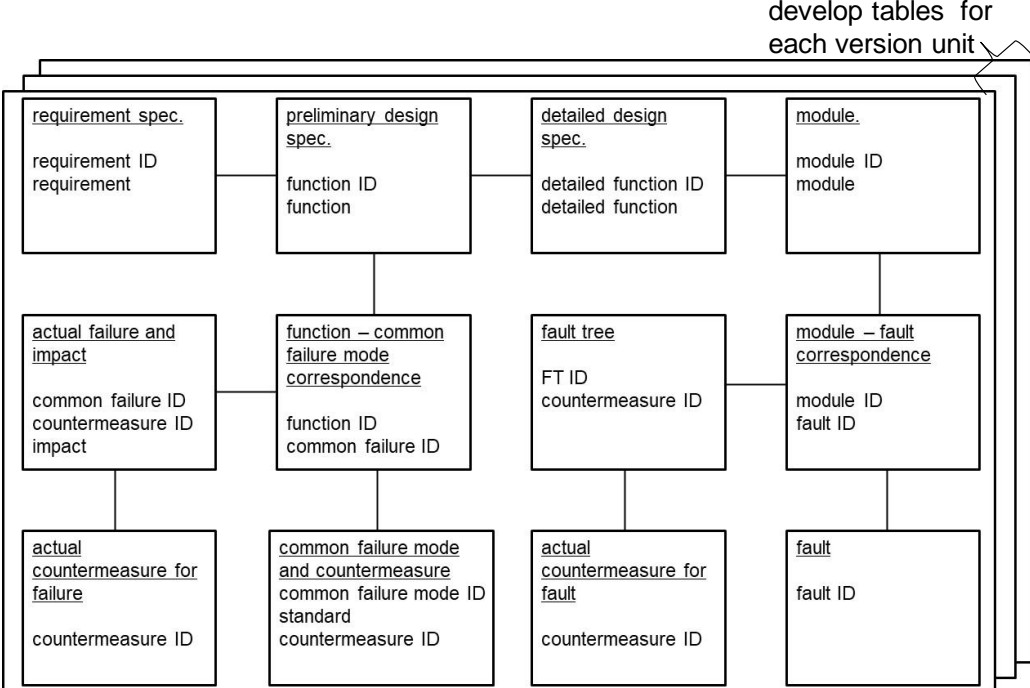

**Figure 9.** Outline of design and safety information database (DSDB) data structure.

## 4. Application and Evaluation

This section describes the application and evaluation of the proposed method. Section 4.1 describes the outline of the target system. Section 4.2 describes the design information of CSW. Section 4.3 describes the result of the safety analysis. Section 4.4 describes the benefits of the proposed method. Section 4.5 describes the list of limitations of the proposed method.

### 4.1. Outline of the Target System

We applied the proposed method to the CSW conducting communication between the control equipment installed into the space station according to the MIL-STD-1553B protocol. The communication between the control equipment is conducted via the data bus. The communication control is managed by BUS-61553B (Device Corporation Inc., Bohemia, NY, USA), and data setting tasks and obtaining tasks are managed by a CPU made by Intel Inc. (Santa Clara, CA, USA) via the shared memory between the BUS-61553B and CPU. Most tasks conducted by BUS-61553B are executed automatically, whereas the tasks conducted by the CPU are conducted by the program written in the C language. The lines of code (LOC) of CSW is approximately 800. BUS-61553B conducts the communication between the controller and the remote terminal (RT). The controller has a master roll, and the RT has a slave roll. The communication starts after the controller turns on the power switch of RT (interrupt of power on). The control signals that RT receives are as follows: interruption of broadcast command (BC) reception, interruption of RT response reception, and interruption of the control cycle (data set, command judgment, and Health and Status (H&S) data creation). When there is an interruption of BC reception, BUS-61553B writes the contents of the received command to the stack and revises the data reference pointer. When there is an interruption of RT response reception, CSW creates and sends the RT data. When there is an interruption of the control cycle, RT investigates the communication status stored in the memory and conducts an adequate task corresponding to the received command variation. The reception frequency of the command is approximately one time per 125 ms (8 Hz), and the cycle of interruptions of the control cycle is 125 ms, which occurs asynchronously.

### 4.2. Design Information on the CSW

Figure 10a–c shows the design result of the communication CSW. Here, we focus on and discuss the contents of process 3 in Figure 10b,c. Generally, SADM describes DFDs, CFDs, and CSPECs in a separate sheet, but we describe this information in the same sheet to save space. Figure 10a explains the DCD and CCD. Figure 10b shows the DFD, CFD, and CSPEC in the second layer. The interruption of BC reception and the interruption of the control cycle (command judgment) activate process 3, and the interruption of RT response reception and the interruption of the control cycle activate process 2. Figure 10c shows the DFD, CFD, and CSPEC related to process 3. When the interruption of BC reception occurs, the processes (3.1, 3.2, 3.3, and 3.7) are executed by the activation table. Those processes' concrete operations are as follows: increment stack pointer, set necessary data to descriptor stack, and register data blocks in the BC command to the memory. Those processes are conducted by BUS-61553B automatically. When the control cycle (command judgment) interruption occurs, the processes (3.4, 3.5, and 3.6) are activated by the activation table. The concrete operations are as follows: extract the descriptor that is referred by the stack pointer from the stack, investigate the status of the received command, and extract data blocks that are referred by the address written in descriptor from memory.

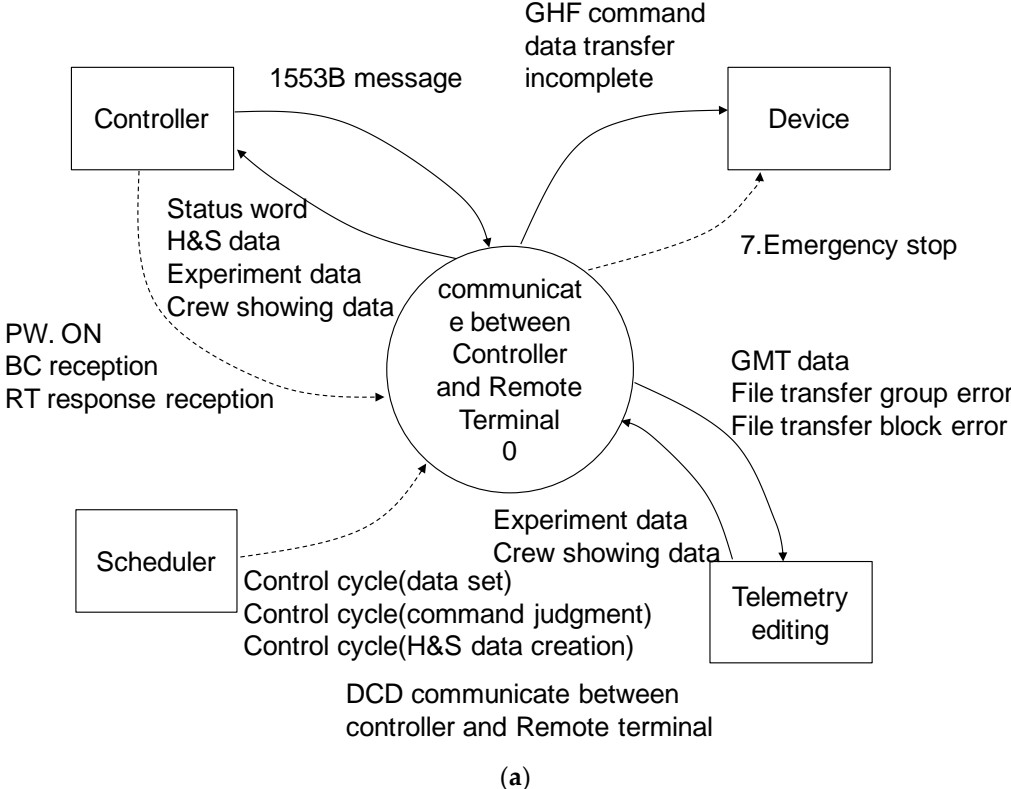

(**a**)

**Figure 10.** *Cont.*

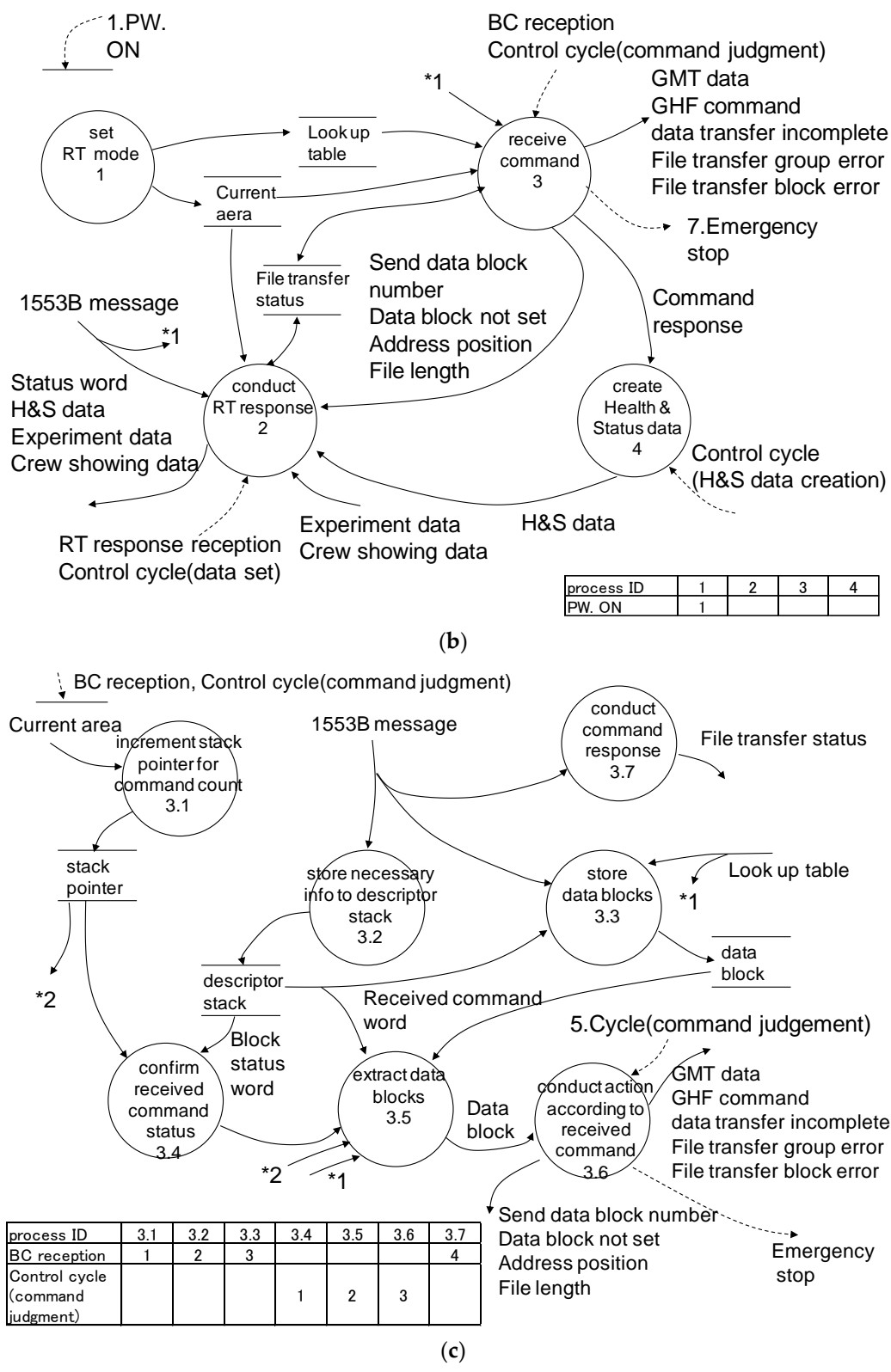

**Figure 10.** (**a**) Design Result of Communication CSW–Data Context Diagram (DCD) and Control Context Diagram (CCD). (**b**) Design Result of Communication CSW–Data Flow Diagram (DFD), Control Flow Diagram (CFD), and Control Specifications (CSPEC) in the Second Layer. (**c**) Design Result of Communication CSW–DFD, CFD, and CSPEC of Process 3.

### 4.3. Safety Analysis Results Related to CSW

This subsection describes the results of the safety analysis to CSW of the communication control equipment. The safety analysis of the existing CSW of the communication control equipment was conducted by the engineers. The engineers who designed this CSW had sufficient experience that the engineers developed more than 10 CSWs of the aerospace systems and had enough knowledge about the SADM and the safety analysis method. The engineer developed the CSW obeying the guideline [39]; while the engineers who conduct safety analysis using the proposed method had the same experience and knowledge and mastered the proposed method additionally.

First, we explain the results of FMEA, which is conducted using the information of DFDs, CFDs, and CSPECs in the second and/or third layer from DSDB. Table 3 shows the results of FMEA conducted to the functions of process 3.X. There are no countermeasures for the processes (3.1, 3.2, 3.3, and 3.7) because BUS-61553B activates those processes automatically (by hardware). In process 3.4, there are no countermeasures that can be added because the format of the data blocks sent and/or received is defined by the hardware and protocol of BUS-61553B. In process 3.5, the reliability of received data blocks can be improved by adding the error-recovering bit. In process 3.6, the operation's reliability corresponding to the received command can be improved by registering the obtained data multiplexed. As a result, we add the following countermeasures: add the error-recovering bit in the data block and register the obtained data multiplexed. The total man-hours in conducting FMEA and planning countermeasures are approximately 20 h. (The man-hours of developing DFD, CFD, and CSPEC, as well as the man-hours of modifying the CSW, are not also included.)

**Table 3.** Result of FMEA.

| Process ID | Function | Common Failure Modes | Causes | Impact to System | Severity | Probability | Detection Rate | Risk | Countermeasures |
|---|---|---|---|---|---|---|---|---|---|
| 3.1 | increment stack pointer for command count (conducted by BUS61553B) | malfunction of hardware | malfunction of BUS61553B | cannot receive commands | High | Low | Low | High | Not Applicable (NA) (use high reliability parts) |
| 3.2 | store necessary info to descriptor stack (conducted by BUS61553B) | malfunction of hardware | malfunction of BUS61553B | cannot receive commands | High | Low | Low | High | NA (use high reliability parts) |
| 3.3 | store data blocks (conducted by BUS61553B) | malfunction of hardware | malfunction of BUS61553B | cannot receive commands | High | Low | Low | High | NA (use high reliability parts) |
| 3.4 | confirm received command status | activation conditions failure | no sync data | cannot receive commands | High | Low | Low | High | multiplexing timer interruption, multiplexing descriptor pointer, multiplexing descriptor pointer (Spec. of BUS61553B) |
| | | completion conditions failure | cannot access descriptor stack | | | | | | |
| | | inadequate input data | error of descriptor stack | | | | | | |
| | | inadequate output data | command reception error on | | | | | | |
| | | inadequate algorithm | NA | NA | - | - | - | - | - |
| | | program destruction | malfunction of memory | cannot receive commands | High | Low | Low | High | multiplexing error bit for command reception |
| | | back up error | NA | NA | - | - | - | - | |
| | | security error | NA | NA | - | - | - | - | - |

**Table 3.** *Cont.*

| Process ID | Function | Common Failure Modes | Causes | Impact to System | Severity | Probability | Detection Rate | Risk | Countermeasures |
|---|---|---|---|---|---|---|---|---|---|
| | | operation procedure miss | NA | NA | - | - | - | - | |
| | | malfunction of hardware | malfunction of Electronic Control Unit (ECU) | NA | - | - | - | - | |
| 3.5 | extract data blocks | activation conditions failure | cannot finish process 3.4 | cannot receive commands | High | Low | Low | High | countermeasure in process 3.4 |
| | | completion conditions failure | cannot access data blocks | cannot receive commands | High | Low | Low | High | multiplexing data block pointer (Spec. of BUS61553B) |
| | | inadequate input data | data block error | cannot conduct action corresponding to received command | Middle | Low | High | Low | increase retry number of sending command (Spec. of 1553B protocol) |
| | | inadequate output data | command reception error | request to send command | Middle | Low | High | Low | |
| | | inadequate algorithm | NA | NA | - | - | - | - | - |
| | | program destruction | malfunction of memory | cannot extract data blocks | High | Low | High | Low | add check bit to data block |
| | | back up error | NA | NA | - | - | - | - | - |
| | | security error | NA | NA | - | - | - | - | - |
| | | operation procedure miss | NA | NA | - | - | - | - | - |
| | | malfunction of hardware | malfunction of ECU | NA | - | - | - | - | - |
| 3.6 | conduct action according to received command | activation conditions failure | cannot finish process 3.5 | cannot response for command | High | Low | Low | High | countermeasure in process 3.5 |
| | | completion conditions failure | NA | NA | - | - | - | - | - |
| | | inadequate input data | cannot conduct adequate command response | cannot response for command | High | Low | Low | High | add command send request function |
| | | inadequate output data | cannot conduct adequate command response | cannot response for command | High | Low | Low | High | add command send request function |
| | | inadequate algorithm | NA | NA | - | - | - | - | - |
| | | program destruction | NA | NA | - | - | - | - | - |
| | | back up error | NA | NA | - | - | - | - | - |
| | | security error | NA | NA | - | - | - | - | - |
| | | operation procedure miss | NA | NA | - | - | - | - | - |
| | | malfunction of hardware | malfunction of ECU | cannot response for command | High | Low | High | Low | use redundant system |
| 3.7 | conduct command response (conducted by BUS61553B) | malfunction of hardware | malfunction of BUS61553B | cannot response for command | - | - | - | - | - |

Next, we describe the result of the FTA. Here, we discuss the FT related to the fault that loses the BC command except for the BC command received immediately before when CSW receives plural BC commands. Figure 11a–d shows the FTs. Figure 11a is the FT corresponding to the operation that pops the descriptors from the stack. We found that

the causes of this fault are as follows: (A) the algorithm of process 3.5 is inadequate, (B) the value of the stack pointer is not the value of the lost BC commands, (C) the value of the descriptor is not the value of the lost BC command, and (D) the obtained data block is not the data block of the lost BC command. Figure 11a–d shows the detailed FT related to causes (A–D), respectively. Figure 11a shows the FT related to cause (A). We then explain the preliminary events obtained by conducting FTA: "the algorithm of process 3.5 is inadequate". As the results of analyzing Figure 11b–d, causes (B–D) are revised by BUS-61553B simultaneously, and those refer to the command received immediately before. As a result of investigating process 3.6, the algorithm obtains (pops) the data block referred to as the descriptor at the top of the stack. This algorithm is adopted because the frequency of the interruption of BC reception and the cycle of the interruption of the control cycle is the same. However, practically those interruptions occur asynchronously. Additionally, there exists the deviation of the frequency of occurrence of the interruption of completion of BC reception. Therefore, when plural receptions of BC within an occurrence of interruption of the control cycle exist, it finds that the data blocks are not obtained (popped), except for BC's data block immediately before. As a result, we clarified the causes (preliminary events) related to BC's loss, except for what the BC received immediately before. We also found a possibility that there will be stack overflow and memory (data area) destruction when CSW is used for the long term because the non-extracted (popped) data blocks remain as the garbage in the stack, the descriptor stack, and data block area. As a result, we proposed the following countermeasures when the interruption of the control cycle occurs: (I) add an operation that extracts (pops) the data blocks until the stack pointers and descriptor stack become empty, (II) add the data escape area that temporarily saves the read data block until it can be processed, and (II) add a flag that shows that there exist data blocks waiting for the conduction. As a result of consideration of the CSW's resources and the modification costs, the countermeasure (III) is only applied to the CSW because the frequency of this fault is much rarer. The total man-hours of creating this FT and planning the countermeasures is approximately 10 h (the man-hours of developing DFD, CFD, and CSPEC, as well as the man-hours of modifying CSW, are not included).

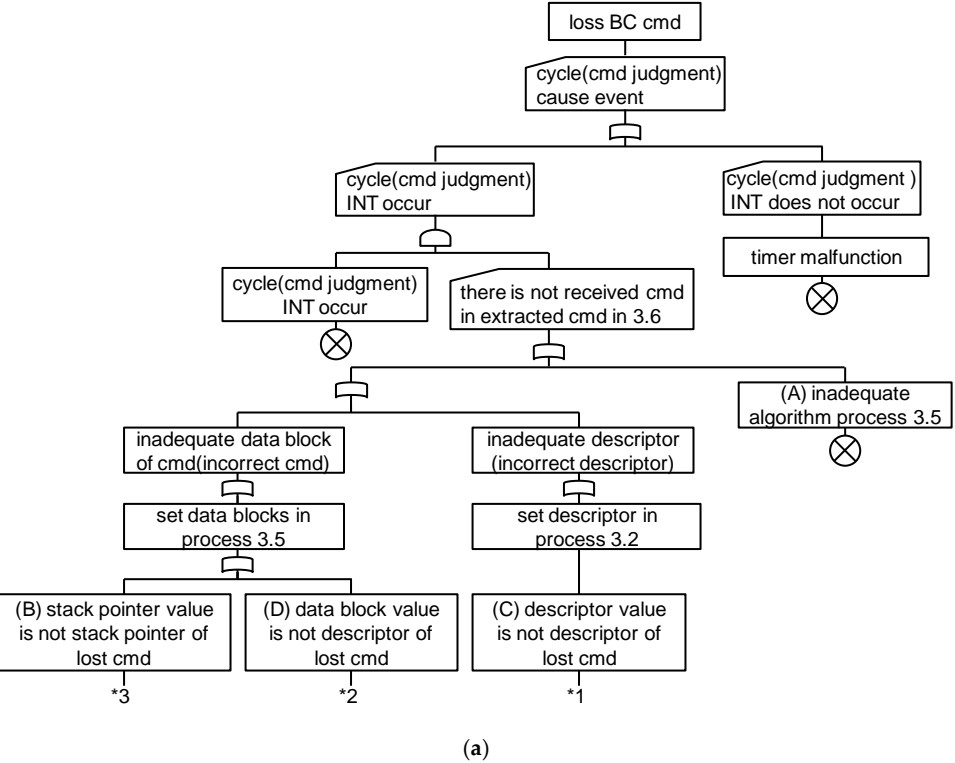

(a)

**Figure 11.** *Cont.*

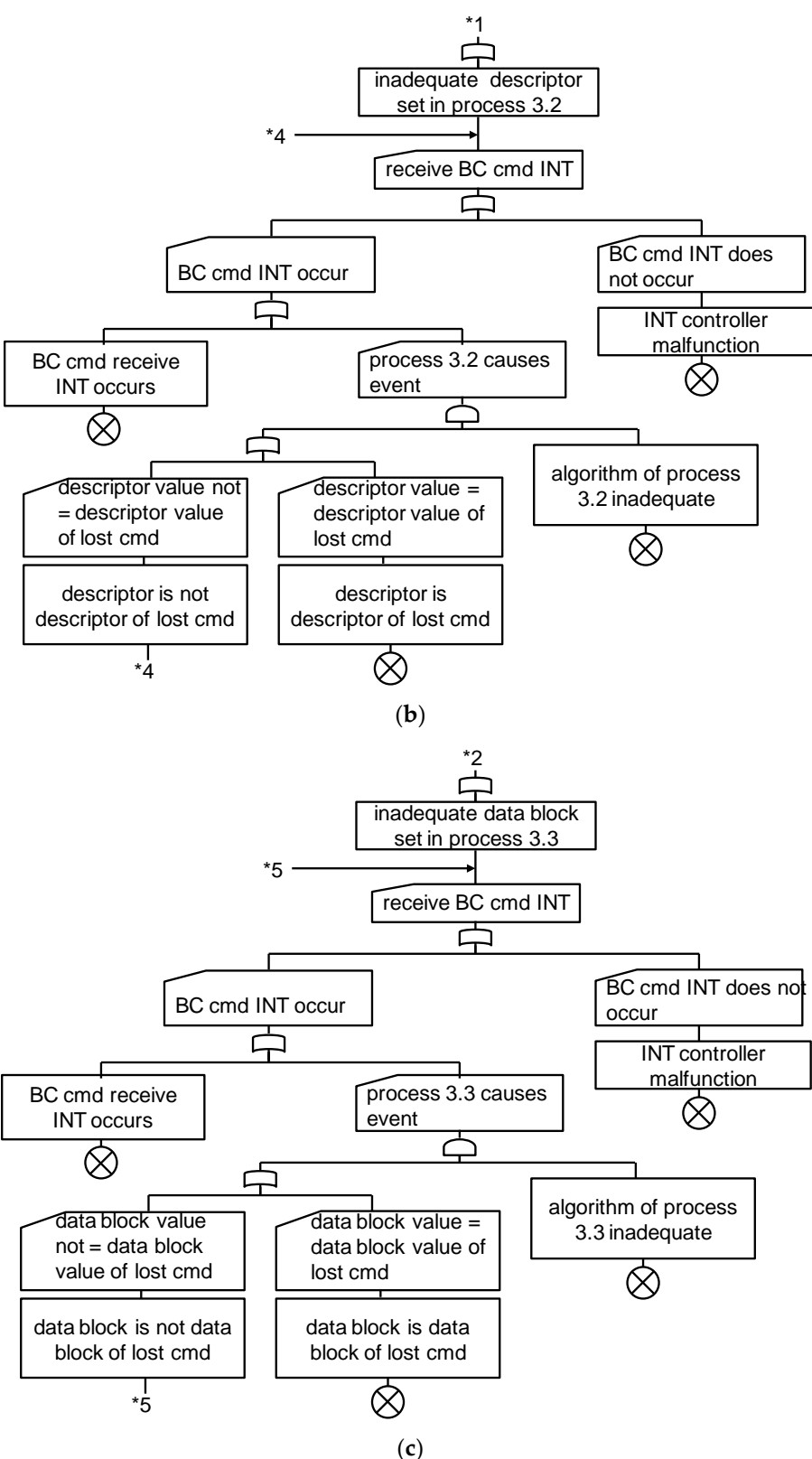

**Figure 11.** *Cont*.

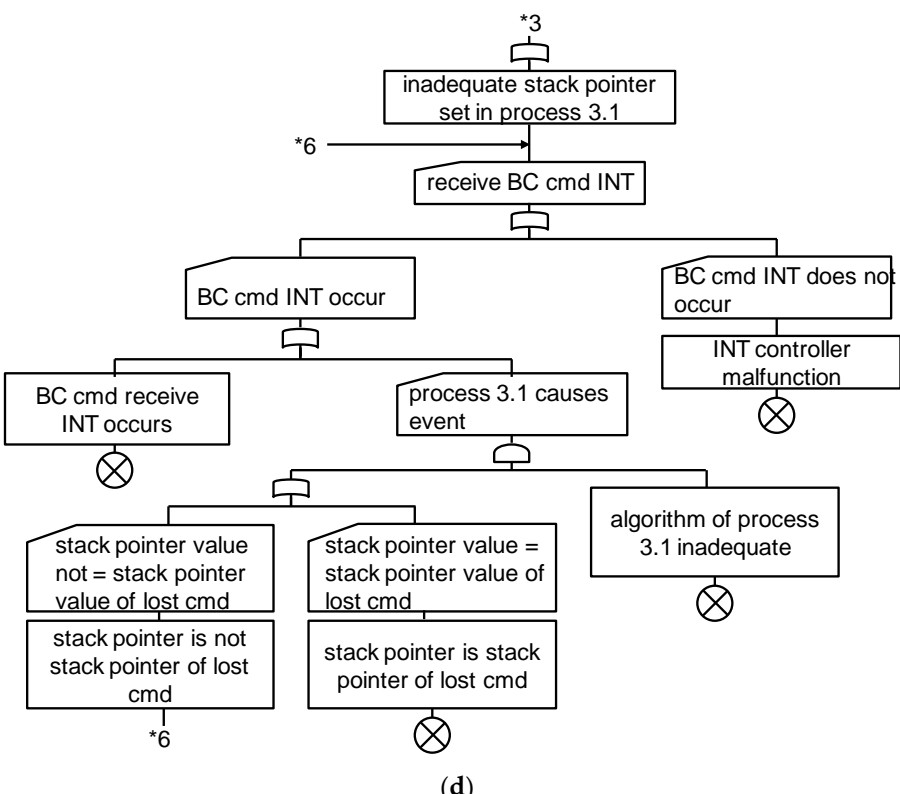

**(d)**

**Figure 11.** (**a**). Developed FT—Outline of FT. (**b**) FT—Detail of (C) Descriptor Value Is Incorrect. (**c**) Developed FT—Detail of (D) Data Block Value Is Incorrect. (**d**) Developed FT—Detail of (B) Stack Pointer Value Is Incorrect.

### 4.4. Benefits of the Proposed Method

First, the benefits of the usage of the proposed FMEA method are described. In conventional FMEA, the definition of a failure mode is different depending on the analyst. Therefore, it could be said that the result of FMEA depended on the analyst. In the proposed FMEA, common failure modes were defined as deviations from the original usage of the function. This made it easy to correspond between the processes in DFD and the functions and to conduct FMEA without depending on the granularity of the function. As a result, the analyst with a certain level of experience and knowledge can obtain the same adequate FMEA results.

Second, the benefits of the usage of the proposed FTA method are described. In conventional FTA, the completeness of the FTA results was different depending on the experience and knowledge of the analyst. Additionally, the well-experienced analyst sometimes omitted the obvious parts of the developed FT. Therefore, when the other analyst investigated the FT developed by the well-experienced analyst, the other analyst sometimes could not understand how the well-experienced analyst came to such a conclusion. In the proposed FTA, the CSW's instruction was defined as the minimum component of the FT, and the FT development rules were defined. As a result, the analyst with a certain level of experience and knowledge became able to obtain the same adequate FTA results.

Third, the whole proposed method is described. In the proposed method, the design information and safety analysis results could be managed unitarily, and those are shared in the whole CSW's development process. As a result, we could conduct designing and safety analysis in pairs in the adequate CSW's development stage and become able to feedback the safety analysis results to the design information. This reduced the additional works related to the implementation of safety countermeasures after the completion of the CSW. Finally, we could develop the CSW with an adequate program structure (CSW without the conflicts between the normal functions and the safety functions). By rigorously applying

the proposed method, the CSW's design and safety analysis have come to be carried out appropriately. As a result, engineers with a certain level of experience and ability can develop CSWs with high quality. Furthermore, by the reduction of backtracking work, a reduction of development cost is expected.

From the above-mentioned results, we can conduct the reproducible safety analysis and develop the adequate CSW that is reflected in the safety analysis results. In the future, we will enrich the variety of the common failure modes and the FT templates by feedbacking the other development results. We consider that the benefits of the usage of the proposed method will increase.

*4.5. List of Limitations*

The proposed method can cooperate with FMEA and FTA by managing CSW's design information and safety analysis results unitarily. In the CSW's upper development process, the comprehensive failure countermeasures can be adopted to the CSW by conducting FMEA, whereas in the CSW's lower development process, the safety analysis and countermeasures for a specific fault can be adopted by conducting FTA. Additionally, safety analysis can be done within a reasonable time. Therefore, the reliability of CSW improves by applying the proposed method. By applying and evaluating the proposed method, we clarified the following problems.

4.5.1. The Issue of the CSW's Size

Recently, the CSW became larger (the scale of avionics software of the newest aircraft has reached 20 million LOC). Because the CSW has many functions, it is difficult to conduct FMEA for all functions in the given development time. When conducting FTA, it is difficult to reverse-trace the instructions because a function consists of codes in many modules (classes and methods). For FMEA, a method that can analyze safety from the upper to lower layers step by step by classifying and dividing the functions hierarchically will be investigated. For FTA, the analysis method for a validated black box module, such as modules completed in single-unit testing and combined testing, will be investigated. Additionally, we investigated the adequate design guideline because it is necessary to conduct adequate function dividing (adequate functional granularity) and good design (low coupling and high cohesion) to realize the methods mentioned above.

4.5.2. The Issue Related to the Conflicts between Countermeasures and the Addition of New Risk

Generally, the safety of the CSW will be improved by repeating safety analysis because the problems that the CSW has have been clarified, and the countermeasures are applied. If additional countermeasures are applied to the CSW portion that has already been applied with countermeasures, conflicts between old countermeasures and new countermeasures will occur. As a result of adding a new countermeasure to a function, there is a possibility of negatively impacting other functions. In this case, the following countermeasures are required: applying only the countermeasure that the function has a greater negative impact and redesigning the CSW that does not have conflicts. We investigated a method to judge whether redesigning or not was based on the risk evaluation results using Figure 3.

4.5.3. The Issue of Attacks

The proposed method improves safety by detecting the problems of the CSW's functions based on good design; however, enough countermeasures for the attack from the outside of CSW have not been considered. In the future, because it is considered that CSWs will become the Internet of things (IoT), we have to investigate security in addition to safety.

### 4.5.4. The Issue of Using an Object-Oriented Programming Language

Recently, there is an increase in cases of CSW developments using an object-oriented programming language. However, it is not easy to apply the proposed method to the CSW developed using an object-oriented programming language. FMEA can apply to those CSWs by corresponding the unit of the function to the classes' methods. Applying FTA to CSWs requires developing a new FT template corresponding to the object-oriented programming language. Additionally, the FT development rules cannot deal with inheritance and polymorphism; thus, we have to investigate applying the proposed method to the object-oriented language.

### 4.5.5. The Issue of Other Safety Analysis Methods

The proposed method adopts FMEA and FTA as safety analysis methods. Recently, accidents that have not been considered before have occurred because of the increasing complexity of the CSW's functions. In the future, the proposed method should adopt other safety analysis methods, such as HAZOP, STPA, and the Functional Resonance Analysis Method (FRAM) [40], to realize a safer CSW.

## 5. Conclusions and Future Works

This paper proposes a method that develops a safer CSW and a safety analysis environment by managing the CSW's design information and safety analysis results unitarily and cooperating with multiple safety analysis methods (FMEA and FTA). Additionally, the proposed method and the environment were applied to the development and safety analysis of the communication CSW installed into control equipment on the space system. It was found that we can plan adequate countermeasures for realizing safety CSW within an adequate analysis time.

The points to be noted when using the proposed method are now described. The proposed method becomes effective by managing design information and safety analysis results unitarily throughout the development process. The organizations need to define the CSW's development standards that adapt to the proposed method and supervise the engineers to follow standards. Additionally, it is necessary to adopt the proposed method to the development standards, since the development process is different for each organization.

We now detail future works. At first, we will apply the proposed method and the environment to more CSWs and improve them by reflecting on the application results. Next, since there are restrictions described in Section 4.5 to using the proposed method, these will have to be solved. It is necessary to propose a safety analysis method for attacks through the Internet because CSW will always connect to the Internet. Additionally, it is necessary to propose a safety analysis method for AI modules (machine learning modules) because the AI modules will be added to the conventional CSW to improve CSW's functions. Additionally, we will investigate the proposed method by adopting new safety analysis methods, because many new analysis methods are proposed, such as HAZOP, STPA, and FRAM, etc.

**Author Contributions:** Conceptualization, M.T.; Discussion, M.T., Y.A. and Y.W.; writing—original draft preparation, M.T.; writing—review and editing, M.T., Y.A. and Y.W.; supervision, M.T.; funding acquisition, M.T. All authors have read and agreed to the published version of the manuscript.

**Funding:** This research was supported by a Grant-in-Aid for Scientific Research (C) of the Japan Society for the Promotion of Science, grant number 19K04920, title, "Integrated analysis method for hazard caused by software interaction cooperating with multiple safety analysis methods".

**Institutional Review Board Statement:** Not applicable.

**Informed Consent Statement:** Not applicable.

**Data Availability Statement:** The part of data presented in this study are available on request from the corresponding author. The data are not publicly available due to limitation of the security for the product.

**Conflicts of Interest:** The authors declare no conflict of interest.

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
