# Peer review of "A Safety Analysis Method for Control Software in Coordination with FMEA and FTA"

_information, doi:10.3390/info12020079_

Round 1

Reviewer 1 Report

This is a good paper summarizing the practice of software reliability design and improvement. The authors may improve the "method" in a more constructed and theoretical manner. Please also improve the technical writing of the manuscript. Some texts in FIGURES are not legible. 

Author Response

Thank you for your important suggestions. The answers for your suggestion are described as follows.

-The authors may improve the method in a more constructed and theoretical manner.

The authors add some explanations and examples into section 3.2 and 3.3 to help the understanding of readers.

-Improve technical writing.

The authors review the manuscript again and use external services to improve the manuscript.

-Some texts in Figures are not legible.

The authors revise the text of Figures and Tables.

Reviewer 2 Report

The paper is interesting, but the quality of the writing is below zero, I suggest to authors first rewrite the paper from first to end, then resubmit to check. For example:

Why a paper should have two introductions? 

Several keywords are not defined.

The figures provided in a very bad structure.

The tables also the same problem.

The novelty of the paper is no clear.

Some figures are not figures, some texts are not figures, I confused.

Why the figures should be 5(1), 5(2).... very confusing.

Several figures provided without any explanation.

Does necessary to have several figures in the paper without any useful information? 

Totally, the paper is too confusing and it is very chaotic, I got a headache, therefore, I decide to reject the paper to rewrite from first to end. 

Author Response

Thank you for your important suggestions. The answers for your suggestion are described as follows.

-Why a paper should have two introductions?

The title of Section 2 was mistake. The author modified it (related works).

-Several Keywords are not defined.

The author checked and revised Keywords.

-The figures provide in a very bad structure.

 The tables also the same problem.

The authors checked and revised Figures and Tables.

-The novelty of the papers is not clear.

The authors checked and revised the manuscript and use the external service.

-Some figures are not figures, some texts are not figures.

The authors checked the figures.

-Why the figures should be 5(1), 5(2)…

The authors considered that figure 5(1)-5(9) were the FT template group. So, we give them the same figure number. Each figure can identify using (x). (x is the number, 1-9.)

-Several figures provided without any explanation.

The authors add explanations. If additional explanations are required, please show the concrete figure numbers. The authors will add the explanations.

-Does necessary to have several figures in the paper without any useful information?

The authors considered that all figures are necessary to write this manuscript. If the reviewer gives us information, the authors will del

Reviewer 3 Report

This paper proposes an approach to analyze control software in a unified way by combining FMEA and FTA. The contribution is a two-layer development method whether FMEA is performed at the upper level, along with planning and requirements, and FTA is performed at the lower level of programming. The FTA is supported by fault tree templates for C language, and also a tool is provided to support the analysis process. The authors evaluate their approach on a control program for a space station, finding that their approach made that program more reliable by identifying faults and adding countermeasures.

The paper is of interest because it takes a new, creative look at the classical analyses and ties them together in a unified, continuous framework. This contribution should be of interest to engineers of many safety-critical software-controller systems.

Another strong point is that the method connects closely to the implementation by the means of fault tree templates for C. So it stands out positively to many other methodologies that operate primarily in the headspace of software engineers and do not explicitly relate to the code.

Finally, the evaluation is convincing because of the concrete faults and countermeasures described for a specific piece of software. This shows that the proposed method can obtain practical benefits. It would be good to see it applied to a substantially different system, but one example conveys a lot already.

The weakest aspect of the manuscript is that it is rather abstract and heavyweight early on, in Section 3. It is difficult to read as a result. The authors should consider adding examples and clarifications to the heavily abstract discussion.

Another weak aspect is the English use: some phrases are awkward or ungrammatical. The text requires substantial editing; the authors are encouraged to use an external service if possible. One particularly misleading and pervasive error is the past tense of the meta-narrative about the paper flow, such as "We explained", whereas it should be "Here we explain" or "Below we will explain" or something similar. The past tense should be reserved for events that happened prior to the reader walking through this paper (e.g., some other method was discovered or the authors evaluated their method).

Minor issues:
- The abstract introduces the abbreviation SADM but never uses it
- "safing CSW" is ungrammatical
- The first paragraph of the intro talks about some failure events but does not back it with references. Either add references or make it clear that you're talking about generalities, not specific individual events (although even generalities should ideally be backed up by references).
- Section 2 name is wrong
- "research" is singular, not plural. ("Researches" is ungrammatical.)
- "method in safer CSW" is ungrammatical
- Figure 1 should probably be moved earlier in the paper, maybe as early as the introduction. Otherwise, it is difficult to interpret the words "upper" and "lower" without visual help.
- "Improper drugs are manufactured" is difficult to interpret without context (i.e., what is manufacturing the drugs and how the fault is related to that)
- "Invocate" is ungrammatical
- Section 4.4 is called "Evaluation..." but really all the evaluation has happened already earlier in section 4. This subsection looks more like a discussion or a list of limitations.
- "malicious attacking" is redundant because attacks are never benevolent.

Author Response

Thank you for your important suggestions. The answers for your suggestion are described as follows.

<>

-Section 3 is difficult to read. Authors should consider to adding the examples.

The authors add some explanations and examples in Section 3.2 and Section 3.3.

Please refer the contents in line 214-236 and line 276-292 (please refer the red colored sentences).

-Improve English use.

The authors review the English, and use external services.

<>

- The abstract introduces the abbreviation SADM but never uses it

The word "SADM" in Abstract is deleted.

- "safing CSW" is ungrammatical

The authors modify the word “safing CSW” to “developing a safer CSW”. Please refer line 31.

- The first paragraph of the intro talks about some failure events but does not back it with references. Either add references or make it clear that you're talking about generalities, not specific individual events (although even generalities should ideally be backed up by references).

The reference [1]-[3] are added. Accordingly, numbers of subsequent references are changed.

- Section 2 name is wrong

It was a mistake. The title of section 2 is changed to "Related Works".

- "research" is singular, not plural. ("Researches" is ungrammatical.)

The word "researches" is modified.

- "method in safer CSW" is ungrammatical

The authors modify the word “method in safer CSW” to “method in developing a safer CSW”. Please refer line 131.

- Figure 1 should probably be moved earlier in the paper, maybe as early as the introduction. Otherwise, it is difficult to interpret the words "upper" and "lower" without visual help.

Figure1 is moved to Section 1. The supplementary explanations for the words "upper processes" and ""lower processes" are added.

- "Improper drugs are manufactured" is difficult to interpret without context (i.e., what is manufacturing the drugs and how the fault is related to that)

The authors revise the Table 1.

- "Invocate" is ungrammatical

The authors modify the word “invocate” to “activate”. (Please refer line 466, 467, 473, and 497.)

- Section 4.4 is called "Evaluation..." but really all the evaluation has happened already earlier in section 4. This subsection looks more like a discussion or a list of limitations.

The title of section 4.4 is modified to "List of limitations".

- "malicious attacking" is redundant because attacks are never benevolent.

"Malicious attacking" is replaced to "attacking".

Round 2

Reviewer 2 Report

Figure 1. should be moved in other sections.

Table 1 and Table 2 should be presented in editable format. 

Still, there are several useless figures in the paper. 

Several keywords are not defined.

explanation of all figures must be added in the paper. 

The novelty of the papers is not clear.

The font of tables, figures and the captions are different. 

Another important weakness of the paper is related to discussion, the authors never discussed about the results of the study and never provided arguments why did you conduct this study, also never discussed about the benefits of the proposed method, also, how practitioners can use the proposed method in the real life problems, how the proposed method is useful for future studies.

The conclusion section is another weakness in this study, the conclusion section is not useful, authors need to conclude their work in-depth, the limitations and recommendations for future studies should be provided in the paper, not only simply other studies can do that, can do this, authors need to discuss about the limitations of the proposed method as well as case study limitations, what are your recommendations for future works, how the proposed method solved the case study problem.

The introduction section is rewritten in very bad structure, the introduction section must highlights the novelty, contributions, motivations and objectives of the paper very clearly, but the authors missed to do that. 

Another weakness is about the literature review, the literature review is not only put some sentences and paragraphs, then you called it literature review, these sentences and paragraphs are in other published books and papers. The literature review must highlights the novelty and contribution of the study, The major defect of this study is the debate or Argument is not clearly stated in the introduction session. I would suggest the author improve your theoretical discussion and arrives in your debate or argument. 

The research method of this paper is not useful.

Sensitivity analysis is an important step in analysis process, but authors never present this analysis.

Comparison of the method is another lack of this paper; authors must compare the results of the analysis with other existing approaches. 

Validation of the proposed method is not provided.

Round 3

Reviewer 2 Report

The paper much better now, there are some grammatical errors in the paper. 
